# Asynchronous Parallel Greedy Coordinate Descent

**Yang You** $^{\diamond,+}$   **XiangRu Lian** $^{\dagger,+}$   **Ji Liu** $^{\dagger}$   **Hsiang-Fu Yu** $^{\ddagger}$
**Inderjit S. Dhillon** $^{\ddagger}$   **James Demmel** $^{\diamond}$   **Cho-Jui Hsieh** $^{*}$
$^{+}$ equally contributed      $^{*}$ University of California, Davis      $^{\dagger}$ University of Rochester
$^{\ddagger}$ University of Texas, Austin      $^{\diamond}$ University of California, Berkeley
youyang@cs.berkeley.edu,  xiangru@yandex.com,  jliu@cs.rochester.edu
{rofuyu,inderjit}@cs.utexas.edu,  demmel@eecs.berkeley.edu
chohsieh@cs.ucdavis.edu

## Abstract

In this paper, we propose and study an Asynchronous parallel Greedy Coordinate
Descent (Asy-GCD) algorithm for minimizing a smooth function with bounded
constraints. At each iteration, workers asynchronously conduct greedy coordinate
descent updates on a block of variables. In the first part of the paper, we analyze the
theoretical behavior of Asy-GCD and prove a linear convergence rate. In the second
part, we develop an efficient kernel SVM solver based on Asy-GCD in the shared
memory multi-core setting. Since our algorithm is fully asynchronous—each core
does not need to idle and wait for the other cores—the resulting algorithm enjoys
good speedup and outperforms existing multi-core kernel SVM solvers including
asynchronous stochastic coordinate descent and multi-core LIBSVM.

## 1 Introduction

Asynchronous parallel optimization has recently become a popular way to speedup machine learning
algorithms using multiple processors. The key idea of asynchronous parallel optimization is to allow
machines work independently without waiting for the synchronization points. It has many successful
applications including linear SVM [13, 19], deep neural networks [7, 15], matrix completion [19, 31],
linear programming [26], and its theoretical behavior has been deeply studied in the past few
years [1, 9, 16].

The most widely used asynchronous optimization algorithms are stochastic gradient method (SG) [7,
9, 19] and coordinate descent (CD) [1, 13, 16], where the workers keep selecting a sample or a
variable randomly and conduct the corresponding update asynchronously. Although these stochastic
algorithms have been studied deeply, in some important machine learning problems a "greedy"
approach can achieve much faster convergence speed. A very famous example is greedy coordinate
descent: instead of randomly choosing a variable, at each iteration the algorithm selects the most
important variable to update. If this selection step can be implemented efficiently, greedy coordinate
descent can often make bigger progress compared with stochastic coordinate descent, leading to a
faster convergence speed. For example, the decomposition method (a variant of greedy coordinate
descent) is widely known as best solver for kernel SVM [14, 21], which is implemented in LIBSVM
and SVMLight. Other successful applications can be found in [8, 11, 29].

In this paper, we study asynchronous greedy coordinate descent algorithm framework. The variable is
partitioned into subsets, and each worker asynchronously conducts greedy coordinate descent in one
of the blocks. To our knowledge, this is the first paper to present a theoretical analysis or practical
applications of this asynchronous parallel algorithm. In the first part of the paper, we formally define
the asynchronous greedy coordinate descent procedure, and prove a linear convergence rate under
mild assumption. In the second part of the paper, we discuss how to apply this algorithm to solve the
kernel SVM problem on multi-core machines. Our algorithm achieves linear speedup with number of
cores, and performs better than other multi-core SVM solvers.

The rest of the paper is outlined as follows. The related work is discussed in Section 2. We propose the asynchronous greedy coordinate descent algorithm in Section 3 and derive the convergence rate in the same section. In Section 4 we show the details how to apply this algorithm for training kernel SVM, and the experimental comparisons are presented in Section 5.

## 2   Related Work

**Coordinate Descent.**   Coordinate descent (CD) has been extensively studied in the optimization community [2], and has become widely used in machine learning. At each iteration, only one variable is chosen and updated while all the other variables remain fixed. CD can be classified into stochastic coordinate descent (SCD), cyclic coordinate descent (CCD) and greedy coordinate descent (GCD) based on their variable selection scheme. In SCD, variables are chosen randomly based on some distribution, and this simple approach has been successfully applied in solving many machine learning problems [10, 25]. The theoretical analysis of SCD has been discussed in [18, 22]. Cyclic coordinate descent updates variables in a cyclic order, and has also been applied to several applications [4, 30].

**Greedy Coordinate Descent (GCD).**The idea of GCD is to select a good, instead of random, coordinate that can yield better reduction of objective function value. This can often be measured by the magnitude of gradient, projected gradient (for constrained minimization) or proximal gradient (for composite minimization). Since the variable is carefully selected, at each iteration GCD can reduce objective function more than SCD or CCD, which leads to faster convergence in practice. Unfortunately, selecting a variable with larger gradient is often time consuming, so one needs to carefully organize the computation to avoid the overhead, and this is often problem dependent. The most famous application of GCD is the decomposition method [14, 21] used in kernel SVM. By exploiting the structure of quadratic programming, selecting the variable with largest gradient magnitude can be done without any overhead; as a result GCD becomes the dominant technique in solving kernel SVM, and is implemented in LIBSVM [5] and SVMLight [14]. There are also other applications of GCD, such as non-negative matrix factorization [11], large-scale linear SVM [29], and [8] proposed an approximate way to select variables in GCD. Recently, [20] proved an improved convergence bound for greedy coordinate descent. We focus on parallelizing the GS-r rule in this paper but our analysis can be potentially extended to the GS-q rule mentioned in that paper.

To the best of our knowledge, the only literature discussing how to parallelize GCD was in [23, 24]. A thread-greedy/block-greedy coordinate descent is a *synchronized* parallel GCD for $L_1$-regularized empirical risk minimization. At an iteration, each thread randomly selects a block of coordinates from a pre-partitioned block partition and proposes the best coordinate from this block along with its increment (i.e., step size). Then all the threads are synchronized to perform the actual update to the variables. However, the method can potentially diverge; indeed, this is mentioned in [23] about the potential divergence when the number of threads is large. [24] establishes sub-linear convergence for this algorithm.

**Asynchronous Parallel Optimization Algorithms.**In a synchronous algorithm each worker conducts local updates, and in the end of each round they have to stop and communicate to get the new parameters. This is not efficient when scaling to large problem due to the curse of last reducer (all the workers have to wait for the slowest one). In contrast, in asynchronous algorithms there is no synchronization point, so the throughput will be much higher than a synchronized system. As a result, many recent work focus on developing asynchronous parallel algorithms for machine learning as well as providing theoretical guarantee for those algorithms [1, 7, 9, 13, 15, 16, 19, 28, 31].

In distributed systems, asynchronous algorithms are often implemented using the concept of parameter servers [7, 15, 28]. In such setting, each machine asynchronously communicates with the server to read or write the parameters. In our experiments, we focus on another multi-core shared memory setting, where multiple cores in a single machine conduct updates independently and asynchronously, and the communication is implicitly done by reading/writing to the parameters stored in the shared memory space. This has been first discussed in [19] for the stochastic gradient method, and recently proposed for parallelizing stochastic coordinate descent [13, 17].

This is the first work proposing an asynchronous *greedy* coordinate decent framework. The closest work to ours is [17] for asynchronous stochastic coordinate descent (ASCD). In their algorithm, each worker asynchronously conducts the following updates: (1) randomly select a variable (2) compute the update and write to memory or server. In our AGCD algorithm, each worker will select the best variable to update in a block, which leads to faster convergence speed. We also compare with ASCD algorithm in the experimental results for solving the kernel SVM problem.

# 3 Asynchronous Greedy Coordinate Descent

We consider the following constrained minimization problem:

$$\min_{x \in \Omega} \quad f(x), \tag{1}$$

where $f$ is convex and smooth, $\Omega \subset \mathbb{R}^N$ is the constraint set, $\Omega = \Omega_1 \times \Omega_2 \times \cdots \times \Omega_N$ and each $\Omega_i, i = 1, 2, \ldots, N$ is a closed subinterval of the real line.

**Notation:** We denote $S$ to be the optimal solution set for (1) and $\mathcal{P}_S(x), \mathcal{P}_\Omega(x)$ to be the Euclidean projection of $x$ onto $S, \Omega$, respectively. We also denote $f^*$ to be the optimal objective function value for (1).

We propose the following Asynchronous parallel Greedy Coordinate Descent (Asy-GCD) for solving (1). Assume $N$ coordinates are divided into $n$ non-overlapping sets $S_1 \cup \ldots \cup S_n$. Let $k$ be the global counter of total number of updates. In Asy-GCD, each processor repeatedly runs the following GCD updates:

- Randomly select a set $S_k \in \{S_1, \ldots, S_n\}$ and pick the coordinate $i_k \in S_k$ where the projected gradient (defined in (2)) has largest absolute value.
- Update the parameter by
$$x_{k+1} \leftarrow \mathcal{P}_\Omega(x_k - \gamma \nabla_{i_k} f(x_k)),$$
  where $\gamma$ is the step size.

Here the projected gradient defined by

$$\nabla_{i_k}^+ f(\hat{x}_k) := x_k - \mathcal{P}_\Omega(x_k - \nabla_{i_k} f(\hat{x}_k)) \tag{2}$$

is a measurement of optimality for each variable, where $\hat{x}_k$ is current point stored in memory used to calculate the update. The processors will run concurrently without synchronization. In order to analyze Asy-GCD, we capture the system-wise global view in Algorithm 1.

---

**Algorithm 1** Asynchronous Parallel Greedy Coordinate Descent (Asy-GCD)

---

**Input:** $x_0 \in \Omega, \gamma, K$
**Output:** $x_{K+1}$
 1: Initialize $k \leftarrow 0$;
 2: **while** $k \leq K$ **do**
 3:     Choose $S_k$ from $\{S_1, \ldots, S_n\}$ with equal probability;
 4:     Pick $i_k = \arg\max_{i \in S_k} \|\nabla_i^+ f(\hat{x})\|$;
 5:     $x_{k+1} \leftarrow \mathcal{P}_\Omega(x_k - \gamma \nabla_{i_k} f(\hat{x}_k))$;
 6:     $k \leftarrow k + 1$;
 7: **end while**

---

The update in the $k^{\text{th}}$ iteration is

$$x_{k+1} \quad \leftarrow \quad \mathcal{P}_\Omega(x_k - \gamma \nabla_{i_k} f(\hat{x}_k)),$$

where $i_k$ is the selected coordinate in $k^{\text{th}}$ iteration, $\hat{x}_k$ is the point used to calculate the gradient and $\nabla_{i_k} f(\hat{x}_k)$ is a zero vector where the $i_k$th coordinate is set to the corresponding coordinate of the gradient of $f$ at $\hat{x}_k$. Note that $\hat{x}_k$ may not be equal to the current value of the optimization variable $x_k$ due to asynchrony. Later in the theoretical analysis we will need to assume $\hat{x}_k$ is close to $x_k$ using the bounded delay assumption.

In the following we prove the convergence behavior of Asy-GCD. We first make some commonly used assumptions:

**Assumption 1.**

    *1. (**Bounded Delay**) There is a set $J(k) \subset \{k - 1, \ldots, k - T\}$ for each iteration $k$ such that*

$$\hat{x}_k \quad := \quad x_k - \sum_{j \in J(k)} (x_{j+1} - x_j), \tag{3}$$

    *where $T$ is the upper bound of the staleness. In this "inconsistent read" model, we assume some of the latest $T$ updates are not yet written back to memory. This is also used in some previous papers [1, 17], and is more general than the "consistent read" model that assumes $\hat{x}_k$ is equal to some previous iterate.*

2. *For simplicity, we assume each set $S_i, i \in \{1, \dots, n\}$ has $m$ coordinates.*
3. *(**Lipschitzian Gradient**) The gradient function of the objective $\nabla f(\cdot)$ is Lipschitzian. That is to say,*

$$\|\nabla f(x) - \nabla f(y)\| \leq L\|x - y\| \quad \forall x, \forall y. \tag{4}$$

*Under the Lipschitzian gradient assumption, we can define three more constants $L_{res}, L_s$ and $L_{max}$. Define $L_{res}$ to be the restricted Lipschitz constant satisfying the following inequality:*

$$\|\nabla f(x) - \nabla f(x + \alpha e_i)\| \leq L_{res}|\alpha|, \quad \forall i \in \{1, 2, ..., N\} \text{ and } t \in \mathbb{R} \text{ with } x, x + te_i \in \Omega \tag{5}$$

*Let $\nabla_i$ be the operator calculating a zero vector where the $i^{th}$ coordinate is set to the $i^{th}$ coordinate of the gradient. Define $L_{(i)}$ for $i \in \{1, 2, \dots, N\}$ as the minimum constant that satisfies:*

$$\|\nabla_i f(x) - \nabla_i f(x + \alpha e_i)\| \leq L_{(i)}|\alpha|. \tag{6}$$

*Define $L_{\max} := \max_{i \in \{1,...,N\}} L_{(i)}$. It can be seen that $L_{max} \leq L_{res} \leq L$.*
*Let $s$ be any positive integer bounded by $N$. Define $L_s$ to be the minimal constant satisfying the following inequality: $\forall x \in \Omega, \forall S \subset \{1, 2, ..., N\}$ where $|S| \leq s$:*

$$\|\nabla f(x) - \nabla f\big(x + \sum_{i \in S} \alpha_i e_i\big)\| \leq L_s \|\sum_{i \in S} \alpha_i e_i\|.$$

4. *(**Global Error Bound**) We assume that our objective $f$ has the following property: when $\gamma = \frac{1}{3L_{\max}}$, there exists a constant $\kappa$ such that*

$$\|x - \mathcal{P}_S(x)\| \leqslant \kappa\|\tilde{x} - x\|, \forall x \in \Omega. \tag{7}$$

*Where $\tilde{x}$ is defined by $argmin_{x' \in \Omega}\left(\langle \nabla f(x), x' - x \rangle + \frac{1}{2\gamma}\|x' - x\|^2\right)$. This is satisfied by strongly convex objectives and some weakly convex objectives. For example, it is proved in [27] that the kernel SVM problem (9) satisfies the global error bound even when the kernel is not strictly positive definite.*

5. *(**Independence**) All random variables in $\{S_k\}_{k=0,1,\dots,K}$ in Algorithm 1 are independent to each other.*

We then have the following convergence result:

**Theorem 2** (Convergence). *Choose $\gamma = 1/(3L_{\max})$ in Algorithm 1. Suppose $n \geq 6$ and that the upper bound for staleness $T$ satisfies the following condition*

$$T(T + 1) \leqslant \frac{\sqrt{n}L_{\max}}{4\mathrm{e}L_{res}}. \tag{8}$$

*Under Assumption 1, we have the following convergence rate for Algorithm 1:*

$$\mathbb{E}(f(x_k) - f^*) \leqslant \left(1 - \frac{2L_{\max}b}{L\kappa^2 n}\right)^k (f(x_0) - f^*).$$

*where $b$ is defined as*

$$b = \left(\frac{L_T^2}{18\sqrt{n}L_{max}L_{res}} + 2\right)^{-1}.$$

This theorem indicates a linear convergence rate under the global error bound and the condition $T^2 \leq O(\sqrt{n})$. Since $T$ is usually proportional to the total number cores involved in the computation, this result suggests that one can have linear speedup as long as the total number of cores is smaller than $O(n^{1/4})$. Note that for $n = N$ Algorithm 1 reduces to the standard asynchronous coordinate descent algorithm (ASCD) and our result is essentially consistent with the one in [17], although they use the optimally strong convexity assumption for $f(\cdot)$. The optimally strong convexity is a similar condition to the global error bound assumption [32].

Here we briefly discuss the constants involved in the convergence rate. Using Gaussian kernel SVM on covtype as a concrete sample, $L_{\max} = 1$ for Gaussian kernel, $L_{res}$ is the maximum norm of columns of kernel matrix ($\approx 3.5$), $L$ is the 2-norm of $Q$ (21.43 for covtype), and conditional number $\kappa \approx 1190$. As number of samples increased, the conditional number $\kappa$ will become a dominant term, and this also appears in the rate of serial greedy coordinate descent. In terms of speedup when increasing number of threads ($T$), although $L_T$ may grow, it only appears in $b = (\frac{L_T^2}{18\sqrt{n}L_{max}L_{res}} + 2)^{-1}$, where the first term inside $b$ is usually small since there is a $\sqrt{n}$ in the demominator. Therefore, $b \approx 2^{-1}$ in most cases, which means the convergence rate does not slow down too much when we increase $T$.

# 4 Application to Multi-core Kernel SVM

In this section, we demonstrate how to apply asynchronous parallel greedy coordinate descent to solve kernel SVM [3, 6]. We follow the conventional notations for kernel SVM, where the variables for the dual form are $\boldsymbol{\alpha} \in \mathbb{R}^n$ (instead of $x$ in the previous section). Given training samples $\{\boldsymbol{a}_i\}_{i=1}^{\ell}$ with corresponding labels $y_i \in \{+1, -1\}$, kernel SVM solves the following quadratic minimization problem:

$$\min_{\boldsymbol{\alpha} \in \mathbb{R}^n} \left\{ \frac{1}{2}\boldsymbol{\alpha}^T Q \boldsymbol{\alpha} - \boldsymbol{e}^T \boldsymbol{\alpha} \right\} := f(\boldsymbol{\alpha}) \quad \text{s.t.} \quad 0 \le \boldsymbol{\alpha} \le C, \tag{9}$$

where $Q$ is an $\ell$ by $\ell$ symmetric matrix with $Q_{ij} = y_i y_j K(\boldsymbol{a}_i, \boldsymbol{a}_j)$ and $K(\boldsymbol{a}_i, \boldsymbol{a}_j)$ is the kernel function. Gaussian kernel is a widely-used kernel function, where $K(\boldsymbol{a}_i, \boldsymbol{a}_j) = e^{-\gamma \|\boldsymbol{a}_i - \boldsymbol{a}_j\|^2}$.

Greedy coordinate descent is the most popular way to solve kernel SVM. In the following, we first introduce greedy coordinate descent for kernel SVM, and then discuss the detailed update rule and implementation issues when applying our proposed Asy-GCD algorithm on multi-core machines.

## 4.1 Kernel SVM and greedy coordinate descent

When we apply coordinate descent to solve the dual form of kernel SVM (9), the one variable update rule for any index $i$ can be computed by:

$$\delta_i^* = P_{[0, \, C]}\big(\alpha_i - \nabla f_i(\boldsymbol{\alpha})/Q_{ii}\big) - \alpha_i \tag{10}$$

where $P_{[0, \, C]}$ is the projection to the interval $[0, \, C]$ and the gradient is $\nabla f_i(\boldsymbol{\alpha}) = (Q\boldsymbol{\alpha})_i - 1$. Note that this update rule is slightly different from (2) by setting the step size to be $\gamma = 1/Q_{ii}$. For quadratic functions this step size leads to faster convergence because $\delta_i^*$ obtained by (10) is the closed form solution of

$$\delta^* = \arg \min_{\delta} f(\boldsymbol{\alpha} + \delta \boldsymbol{e}_i),$$

and $\boldsymbol{e}_i$ is the $i$-th indicator vector.

As in Algorithm 1, we choose the best coordinate based on the magnitude of projected gradient. In this case, by definition

$$\nabla_i^+ f(\boldsymbol{\alpha}) = \alpha_i - P_{[0, \, C]}\big(\alpha_i - \nabla_i f(\boldsymbol{\alpha})\big). \tag{11}$$

The success of GCD in solving kernel SVM is mainly due to the maintenance of the gradient

$$\boldsymbol{g} := \nabla_i f(\boldsymbol{\alpha}) = (Q\boldsymbol{\alpha}) - 1.$$

Consider the update rule (10): it requires $O(\ell)$ time to compute $(Q\boldsymbol{\alpha})_i$, which is the cost for stochastic coordinate descent or cyclic coordinate descent. However, in the following we show that GCD has the same time complexity per update by using the trick of maintaining $\boldsymbol{g}$ during the whole procedure. If $\boldsymbol{g}$ is available in memory, each element of the projected gradient (11) can be computed in $O(1)$ time, so selecting the variable with maximum magnitude of projected gradient only costs $O(\ell)$ time. The single variable update (10) can be computed in $O(1)$ time. After the update $\alpha_i \leftarrow \alpha_i + \delta$, the $\boldsymbol{g}$ has to be updated by $\boldsymbol{g} \leftarrow \boldsymbol{g} + \delta \boldsymbol{q}_i$, where $\boldsymbol{q}_i$ is the $i$-th column of $Q$. This also costs $O(\ell)$ time. Therefore, each GCD update only costs $O(\ell)$ using this trick of maintaining $\boldsymbol{g}$.

Therefore, for solving kernel SVM, GCD is faster than SCD and CCD since there is no additional cost for selecting the best variable to update. Note that in the above discussion we assume $Q$ can be stored in memory. Unfortunately, this is not the case for large scale problems because $Q$ is an $\ell$ by $\ell$ dense matrix, where $\ell$ can be millions. We will discuss how to deal with this issue in Section 4.3.

With the trick of maintaining $\boldsymbol{g} = Q\boldsymbol{\alpha} - 1$, the GCD for solving (9) can be summarized in Algorithm 2.

---

**Algorithm 2** Greedy Coordinate Descent (GCD) for Dual Kernel SVM

---

1: Initial $\boldsymbol{g} = -1, \boldsymbol{\alpha} = 0$
2: For $k = 1, 2, \cdots$
3:     step 1: Pick $i = \arg \max_i |\nabla_i^+ f(\boldsymbol{\alpha})|$ using $\boldsymbol{g}$         (See eq (11))
4:     step 2: Compute $\delta_i^*$ by eq (10)
5:     step 3: $\boldsymbol{g} \leftarrow \boldsymbol{g} + \delta^* \boldsymbol{q}_i$
6:     step 4: $\alpha_i \leftarrow \alpha_i + \delta^*$

---

## 4.2 Asynchronous greedy coordinate descent

When we have $n$ threads in a multi-core shared memory machine, and the dual variables (or corresponding training samples) are partitioned into the same number of blocks:

$$S_1 \cup S_2 \cup \cdots \cup S_n = \{1, 2, \cdots, \ell\} \quad \text{and} \quad S_i \cap S_j = \phi \text{ for all } i, j.$$

Now we apply Asy-GCD algorithm to solve (9). For better memory allocation of kernel cache (see Section 4.3), we bind each thread to a partition. The behavior of our algorithm still follows Asy-GCD because the sequence of updates are asynchronously random. The algorithm is summarized in Algorithm 3.

---

**Algorithm 3** Asy-GCD for Dual Kernel SVM

---

1: Initial $\boldsymbol{g} = -\boldsymbol{1}$, $\boldsymbol{\alpha} = \boldsymbol{0}$
2: Each thread $t$ repeatedly performs the following updates *in parallel*:
3:     step 1: Pick $i = \arg\max_{i \in S_t} |\nabla_i^+ f(\boldsymbol{\alpha})|$ using $\boldsymbol{g}$            (See eq (11))
4:     step 2: Compute $\delta_i^*$ by eq (10)
5:     step 3: For $j = 1, 2, \cdots, \ell$
6:         $g_j \leftarrow g_j + \delta^* Q_{j,i}$ **using atomic update**
7:     step 4: $\alpha_i \leftarrow \alpha_i + \delta^*$

---

Note that each thread will read the $\ell$-dimensional vector $\boldsymbol{g}$ in step 2 and update $\boldsymbol{g}$ in step 3 in the shared memory. For the read, we do not use any atomic operations. For the writes, we maintain the correctness of $\boldsymbol{g}$ by atomic writes, otherwise some updates to $\boldsymbol{g}$ might be overwritten by others, and the algorithm cannot converge to the optimal solution. Theorem 2, suggests a linear convergence rate of our algorithm, and in the experimental results we will see the algorithm is much faster than the widely used Asynchronous Stochastic Coordinate Descent (Asy-SCD) algorithm [17].

## 4.3 Implementation Issues

In addition to the main algorithm, there are some practical issues we need to handle in order to make Asy-GCD algorithm scales to large-scale kernel SVM problems. Here we discuss these implementation issues.

**Kernel Caching.**The main difficulty for scaling kernel SVM to large dataset is the memory requirement for storing the $Q$ matrix, which takes $O(\ell^2)$ memory. In the GCD algorithm, step 2 (see eq (10)) requires a diagonal element of $Q$, which can be pre-computed and stored in memory. However, the main difficulty is to conduct step 3, where a column of $Q$ (denoted by $\boldsymbol{q}_i$)is needed. If $\boldsymbol{q}_i$ is in the memory, the algorithm only takes $O(\ell)$ time; however, if $\boldsymbol{q}_i$ is not in the memory, re-computing it from scratch takes $O(dn)$ time. As a result, how to maintain most important columns of $Q$ in memory is an important implementation issues in SVM software.

In LIBSVM, the user can specify the size of memory they want to use for storing columns of $Q$. The columns of $Q$ are stored in a linked-list in the memory, and when memory space is not enough the Least Recent Used column will be kicked out (LRU technique).

In our implementation, instead of sharing the same LRU for all the cores, we create an individual LRU for each core, and make the memory space used by a core in a contiguous memory space. As a result, remote memory access will happen less in the NUMA system when there are more than 1 CPU in the same computer. Using this technique, our algorithm is able to scale up in a multi-socket machine (see Figure 2).

**Variable Partitioning.**The theory of Asy-GCD allows any non-overlapping partition of the dual variables. However, we observe a better partition that minimizes the between-cluster connections can often lead to faster convergence. This idea has been used in a divide-and-conquer SVM algorithm [12], and we use the same idea to obtain the partition. More specifically, we partition the data by running kmeans algorithm on a subset of 20000 training samples to obtain cluster centers $\{\boldsymbol{c}_r\}_{r=1}^n$, and then assign each $i$ to the nearest center: $\pi(i) = \arg\min_r \|\boldsymbol{c}_r - \boldsymbol{x}_i\|$. This steps can be easily parallelized, and costs less than 3 seconds in all the datasets used in the experiments. Note that we include this kmeans time in all our experimental comparisons.

## 5 Experimental Results

We conduct experiments to show that the proposed method Asy-GCD achieves good speedup in parallelizing kernel SVM in multi-core systems. We consider three datasets: ijcnn1, covtype and webspam (see Table 1 for detailed information). We follow the parameter settings in [12], where $C$

Table 1: Data statistics. $\ell$ is number of training samples, $d$ is dimensionality, $\ell_t$ is number of testing samples.

|         | $\ell$  | $\ell_t$ | $d$ | $C$ | $\gamma$ |
|---------|---------|----------|-----|-----|----------|
| ijcnn1  | 49,990  | 91,701   | 22  | 32  | 2        |
| covtype | 464,810 | 116,202  | 54  | 32  | 32       |
| webspam | 280,000 | 70,000   | 254 | 8   | 32       |

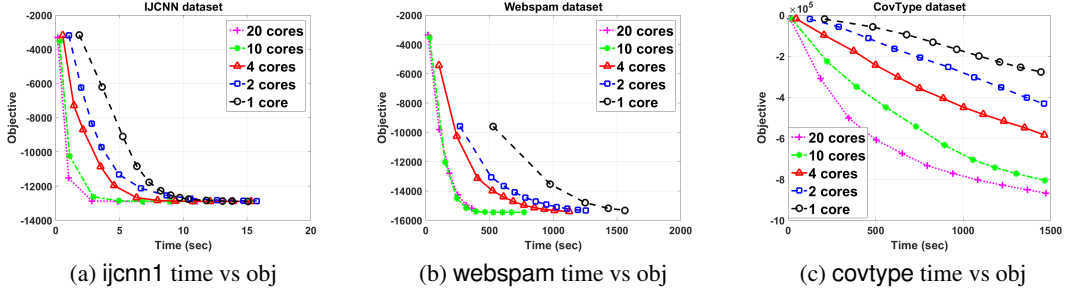

(a) ijcnn1 time vs obj     (b) webspam time vs obj     (c) covtype time vs obj

Figure 1: Comparison of Asy-GCD with 1–20 threads on ijcnn1, covtype and webspam datasets.

and $\gamma$ are selected by cross validation. All the experiments are run on the same system with 20 CPUs and 256GB memory, where the CPU has two sockets, each with 10 cores. We locate 64GB for kernel caching for all the algorithms. In our algorithm, the 64GB will distribute to each core; for example, for Asy-GCD with 20 cores, each core will have 3.2GB kernel cache.

We include the following algorithms/implementations into our comparison:

1. Asy-GCD: Our proposed method implemented by C++ with OpenMP. Note that the preprocessing time for computing the partition is included in all the timing results.
2. PSCD: We implement the asynchronous stochastic coordinate descent [17] approach for solving kernel SVM. Instead of forming the whole kernel matrix in the beginning (which cannot scale to all the dataset we are using), we use the same kernel caching technique as Asy-GCD to scale up PSCD.
3. LIBSVM (OMP): In LIBSVM, there is an option to speedup the algorithm in multi-core environment using OpenMP (see http://www.csie.ntu.edu.tw/~cjlin/libsvm/faq.html#f432). This approach uses multiple cores when computing a column of kernel matrix ($q_i$ used in step 3 of Algorithm 2).

All the implementations are modified from LIBSVM (e.g., they share the similar LRU cache class), so the comparison is very fair. We conduct the following two sets of experiments. Note that another recent proposed DC-SVM solver [12] is currently not parallelizable; however, since it is a meta algorithm and requires training a series of SVM problems, our algorithm can be naturally served as a building block of DC-SVM.

## 5.1 Scaling with number of cores

In the first set of experiments, we test the speedup of our algorithm with varying number of cores. The results are presented in Figure 1 and Figure 2. We have the following observations:

- **Time vs obj (for 1, 2, 4, 10, 20 cores).** From Fig. 1 (a)-(c), we observe that when we use more CPU cores, the objective decreases faster.
- **Cores vs speedup.** From Fig. 2, we can observe that we got good strong scaling when we increase the number of threads. Note that our computer has two sockets, each with 10 cores, and our algorithm can often achieve 13-15 times speedup. This suggests our algorithm can scale to multiple sockets in a Non-Uniform Memory Access (NUMA) system. Previous asynchronous parallel algorithms such as HogWild [19] or PASSCoDe [13] often struggle when scaling to multiple sockets.

## 5.2 Comparison with other methods

Now we compare the efficiency of our proposed algorithm with other multi-core parallel kernel SVM solvers on real datasets in Figure 3. All the algorithms in this comparison are using 20 cores and 64GB memory space for kernel caching. Note that LIBSVM is solving the kernel SVM problem with the bias term, so the objective function value is not showing in the figures.

We have the following observations:

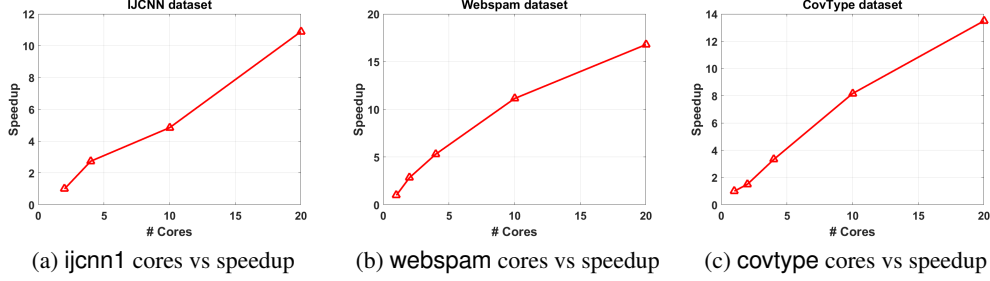

(a) ijcnn1 cores vs speedup     (b) webspam cores vs speedup     (c) covtype cores vs speedup

Figure 2: The scalability of Asy-GCD with up to 20 threads.

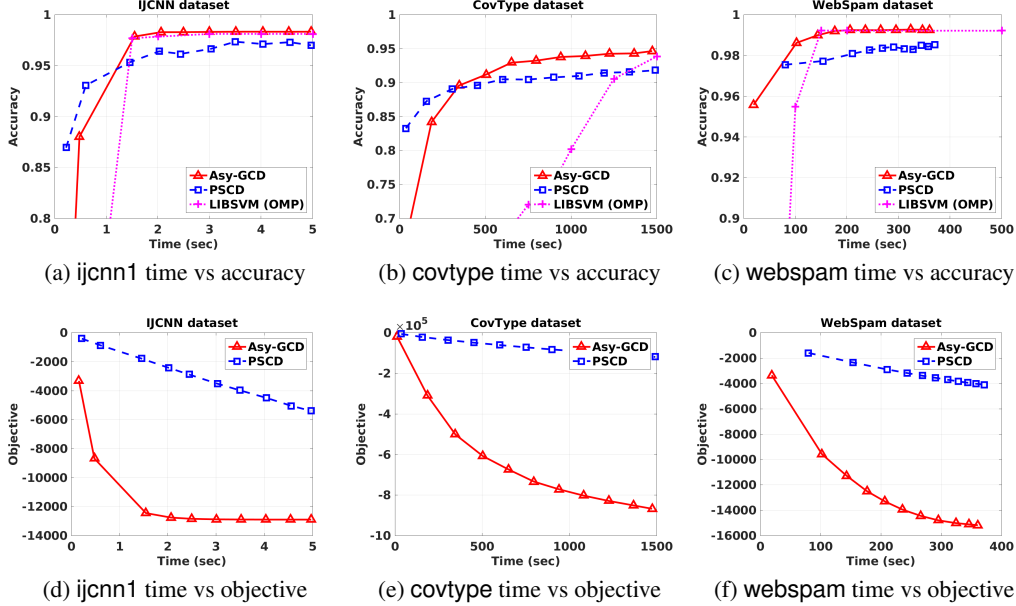

(a) ijcnn1 time vs accuracy     (b) covtype time vs accuracy     (c) webspam time vs accuracy

(d) ijcnn1 time vs objective     (e) covtype time vs objective     (f) webspam time vs objective

Figure 3: Comparison among multi-core kernel SVM solvers. All the solvers use 20 cores and the same amount of memory.

- Our algorithm achieves much faster convergence in terms of objective function value compared with PSCD. This is not surprising because using the trick of maintaining $g$ (see details in Section 4) greedy approach can select the best variable to update, while stochastic approach just chooses variables randomly. In terms of accuracy, PSCD is sometimes good in the beginning, but converges very slowly to the best accuracy. For example, in covtype data the accuracy of PSCD remains 93% after 4000 seconds, while our algorithm can achieve 95% accuracy after 1500 seconds.
- LIBSVM (OMP) is slower than our method. The main reason is that they only use multiple cores when computing kernel values, so the computational power is wasted when the column of kernel ($q_i$) needed is available in memory.

**Conclusions** In this paper, we propose an Asynchronous parallel Greedy Coordinate Descent (Asy-GCD) algorithm, and prove a linear convergence rate under mild condition. We show our algorithm is useful for parallelizing the greedy coordinate descent method for solving kernel SVM, and the resulting algorithm is much faster than existing multi-core SVM solvers.

**Acknowledgement** XL and JL are supported by the NSF grant CNS-1548078. HFY and ISD are supported by the NSF grants CCF-1320746, IIS-1546459 and CCF-1564000. YY and JD are supported by the U.S. Department of Energy Office of Science, Office of Advanced Scientific Computing Research, Applied Mathematics program under Award Number DE-SC0010200; by the U.S. Department of Energy Office of Science, Office of Advanced Scientific Computing Research under Award Numbers DE-SC0008700 and AC02-05CH11231; by DARPA Award Number HR0011-12-2-0016, Intel, Google, HP, Huawei, LGE, Nokia, NVIDIA, Oracle and S Samsung, Mathworks and Cray. CJH also thank the XSEDE and Nvidia support.

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
