[Supplementary Material · nips_asyn_ksvm_supp.pdf]

# A  Appendix

**Definition 1.**

1. *We define following quantities:*

$$
\begin{aligned}
\tilde{x} &:= \arg\min_{x'\in\Omega}\langle\nabla f(x), x'-x\rangle + \frac{1}{2\gamma}\|x'-x\|^2, \\
\bar{x}_k &:= \arg\min_{x\in\Omega}\langle\nabla f(\hat{x}_k), x-x_k\rangle + \frac{1}{2\gamma}\|x-x_k\|^2, \\
\tilde{x}_k &:= \arg\min_{x\in\Omega}\langle\nabla f(x_k), x-x_k\rangle + \frac{1}{2\gamma}\|x-x_k\|^2.
\end{aligned}
$$

2. *For each iteration $k$ we define $A_k$ as the set containing the coordinates should be selected in each block if the cooresponding block is selected in that iteration. Therefore, $A_k = \{a_{k_1}, \ldots, a_{k_n}\}$ where $a_{k_i} \in S_i$ is the selected index of coordinate if $S_i$ is selected in that iteration.*

**Lemma 1.** *Let $\rho$ be a constant with*

$$\rho > \left(1 - 2/\sqrt{n}\right)^{-1} \tag{12}$$

*and define the quantity $\psi$ as follows:*

$$\psi := L_{\max} + \frac{L_{res}T\rho^T}{\sqrt{n}}\left(2 + \frac{L_{\max}}{\sqrt{n}L_{res}} + \frac{2T}{n}\right).$$

*Suppose that the steplength parameter $\gamma > 0$ satisfies the following two upper bounds:*

$$\gamma \leqslant 1/\psi \tag{13}$$

$$\gamma \leqslant \left(1 - \frac{1}{\rho} - \frac{2}{\sqrt{n}}\right)\frac{\sqrt{n}}{4L_{res}T\rho^T}. \tag{14}$$

*Then we have*

$$\mathbb{E}\|x_{k-1} - \bar{x}_k\|^2 \leqslant \rho\mathbb{E}\|x_k - \bar{x}_{k+1}\|^2, k = 1, 2, \ldots.$$

*Proof.* This can be proved by following exactly the same procedure in Theorem 4 of [17] and noting that

$$\|(x_k - \bar{x}_k)_{A_k}\|^2 \leqslant \|x_k - \bar{x}_k\|^2,$$

where $(x_k - \bar{x}_k)_{A_k}$ is a zero vector with coordinates in $A_k$ set to the corresponding coordinates in $x_k - \bar{x}_k$. $\qquad\square$

**Lemma 2.** *Given the requirements in Lemma 1 hold, we have*

$$\mathbb{E}\|\bar{x}_k - x_k\|^2 \geqslant \frac{1}{2\left(\frac{\gamma^2 L_T^2 T^2 \rho^T}{n} + 1\right)}\mathbb{E}\|\tilde{x}_k - x_k\|^2. \tag{15}$$

*Proof.* Since the projection operator is nonexpansive,

$$
\begin{aligned}
\|\bar{x}_k - \tilde{x}_k\| &\leqslant \|P_\Omega(x_k - \gamma\nabla f(\hat{x}_k)) - P_\Omega(x_k - \gamma\nabla f(x_k))\| \\
&\leqslant \|x_k - \gamma\nabla f(\hat{x}_k) - (x_k - \gamma\nabla f(x_k))\| \\
&= \gamma\|\nabla f(\hat{x}_k) - \nabla f(x_k)\|.
\end{aligned}
$$

Thus

$$
\begin{aligned}
\mathbb{E}\|\bar{x}_k - \tilde{x}_k\|^2 \quad &\leqslant \quad \gamma^2 \mathbb{E}\|\nabla f(\hat{x}_k) - \nabla f(x_k)\|^2 \\
&\leqslant \quad \gamma^2 L_T^2 \mathbb{E}\|\hat{x}_k - x_k\|^2 \\
&\overset{(3)}{=} \quad \gamma^2 L_T^2 \mathbb{E}\left\|\sum_{j \in J(k)} (x_{j+1} - x_j)\right\|^2 \\
&\leqslant \quad \gamma^2 L_T^2 T \sum_{j \in J(k)} \mathbb{E}\|x_{j+1} - x_j\|^2 \\
&\leqslant \quad \frac{\gamma^2 L_T^2 T}{n} \sum_{j \in J(k)} \mathbb{E}\|\bar{x}_j - x_j\|^2 \\
&\leqslant \quad \frac{\gamma^2 L_T^2 T^2 \rho^T}{n} \mathbb{E}\|\bar{x}_k - x_k\|^2. \qquad (16)
\end{aligned}
$$

Thus

$$
\begin{aligned}
\mathbb{E}\|\tilde{x}_k - x_k\|^2 \quad &= \quad \mathbb{E}\|\tilde{x}_k - \bar{x}_k + \bar{x}_k - x_k\|^2 \\
&\leqslant \quad 2\mathbb{E}\|\tilde{x}_k - \bar{x}_k\|^2 + 2\mathbb{E}\|\bar{x}_k - x_k\|^2 \\
&\overset{(16)}{\leqslant} \quad \frac{2\gamma^2 L_T^2 T^2 \rho^T}{n} \mathbb{E}\|\bar{x}_k - x_k\|^2 + 2\mathbb{E}\|\bar{x}_k - x_k\|^2 \\
&= \quad 2\left(\frac{\gamma^2 L_T^2 T^2 \rho^T}{n} + 1\right)\mathbb{E}\|\bar{x}_k - x_k\|^2.
\end{aligned}
$$

It immediately follows that

$$
\mathbb{E}\|\bar{x}_k - x_k\|^2 \quad \geqslant \quad \frac{1}{2\left(\frac{\gamma^2 L_T^2 T^2 \rho^T}{n} + 1\right)} \mathbb{E}\|\tilde{x}_k - x_k\|^2.
$$

$\square$

**Proof to Theorem 2**

*Proof.* We start from

$$
\mathbb{E}(f(x_{k+1})) - f^* \quad = \quad \mathbb{E}(f(x_k)) - f^* + \underbrace{\left(\mathbb{E}(f(x_{k+1})) - \mathbb{E}(f(x_k))\right)}_{T_1}.
$$

Firstly, we bound $T_1$ by

$$
\begin{aligned}
T_1 \quad &= \quad \mathbb{E}(f(x_{k+1})) - \mathbb{E}(f(x_k)) \\
&\leqslant \quad \mathbb{E}\langle \nabla f(x_k), x_{k+1} - x_k \rangle + \frac{L_{\max}}{2}\mathbb{E}\|x_{k+1} - x_k\|^2 \\
&= \quad \mathbb{E}\langle \nabla f(\hat{x}_k), x_{k+1} - x_k \rangle + \frac{L_{\max}}{2}\mathbb{E}\|x_{k+1} - x_k\|^2 + \mathbb{E}\langle \nabla f(x_k) - \nabla f(\hat{x}_k), x_{k+1} - x_k \rangle \\
&\leqslant \quad \mathbb{E}\langle \nabla f(\hat{x}_k), x_{k+1} - x_k \rangle + \frac{3L_{\max}}{2}\mathbb{E}\|x_{k+1} - x_k\|^2 \\
&\quad + \mathbb{E}\langle \nabla f(x_k) - \nabla f(\hat{x}_k), x_{k+1} - x_k \rangle.
\end{aligned}
$$

Since

$$
\begin{aligned}
x_{k+1} \quad &= \quad \arg\min_{x \in \Omega_{i_k}} \langle \nabla_{i_k} f(\hat{x}_k), x - x_k \rangle + \frac{1}{2\gamma}\|x - x_k\|^2 \\
&= \quad \arg\min_{x \in \Omega_{i_k}} \langle \nabla_{i_k} f(\hat{x}_k), x - x_k \rangle + \frac{3L_{\max}}{2}\|x - x_k\|^2,
\end{aligned}
$$

and denote $\Omega_{S_k}$ as the Cartesan product of $\Omega_i, i \in S_k$, we have

$$
\begin{aligned}
T_1 & \leqslant \mathbb{E}\left(\min_{x \in \Omega_{i_k}}\left(\langle \nabla f(\hat{x}_k), x - x_k\rangle + \frac{3L_{\max}}{2}\|x - x_k\|^2\right)\right) \\
& \quad +\mathbb{E}\langle \nabla f(x_k) - \nabla f(\hat{x}_k), x_{k+1} - x_k\rangle \\
& \leqslant \mathbb{E}\left(\frac{1}{m}\min_{x \in \Omega_{S_k}}\left(\langle \nabla f(\hat{x}_k), x - x_k\rangle + \frac{3L_{\max}}{2}\|x - x_k\|^2\right)\right) \\
& \quad +\mathbb{E}\langle \nabla f(x_k) - \nabla f(\hat{x}_k), x_{k+1} - x_k\rangle \\
& = \mathbb{E}\left(\frac{1}{mn}\min_{x \in \Omega}\left(\langle \nabla f(\hat{x}_k), x - x_k\rangle + \frac{3L_{\max}}{2}\|x - x_k\|^2\right)\right) \\
& \quad +\mathbb{E}\langle \nabla f(x_k) - \nabla f(\hat{x}_k), x_{k+1} - x_k\rangle.
\end{aligned}
$$

Since $mn = N$, we have

$$
\begin{aligned}
T_1 & \leqslant \frac{1}{N}\mathbb{E}\left(\langle \nabla f(\hat{x}_k), \bar{x}_k - x_k\rangle + \frac{3L_{\max}}{2}\|\bar{x}_k - x_k\|^2\right) \\
& \quad +\frac{1}{n}\mathbb{E}\langle \nabla f(x_k) - \nabla f(\hat{x}_k), (\bar{x}_k - x_k)_{A_k}\rangle \\
& \leqslant \frac{1}{N}\mathbb{E}\left(\langle \nabla f(\hat{x}_k), \bar{x}_k - x_k\rangle + \frac{3L_{\max}}{2}\|\bar{x}_k - x_k\|^2\right) \\
& \quad +\frac{1}{n}\mathbb{E}\|(\nabla f(x_k) - \nabla f(\hat{x}_k))\|\|(\bar{x}_k - x_k)_{A_k}\| \\
& \leqslant \frac{1}{N}\mathbb{E}\left(\langle \nabla f(\hat{x}_k), \bar{x}_k - x_k\rangle + \frac{3L_{\max}}{2}\|\bar{x}_k - x_k\|^2\right) \\
& \quad +\frac{L_{\text{res}}}{n}\mathbb{E}\sum_{j \in J(k)}\|x_{j+1} - x_j\|\|\bar{x}_k - x_k\| \\
& = \frac{1}{N}\mathbb{E}\left(\langle \nabla f(\hat{x}_k), \bar{x}_k - x_k\rangle + \frac{3L_{\max}}{2}\|\bar{x}_k - x_k\|^2\right) \\
& \quad +\frac{L_{\text{res}}}{2n}\mathbb{E}\sum_{j \in J(k)}\left(\alpha\|x_{j+1} - x_j\|^2 + \frac{1}{\alpha}\|\bar{x}_k - x_k\|^2\right) \\
& \leqslant \frac{1}{N}\mathbb{E}\left(\langle \nabla f(\hat{x}_k), \bar{x}_k - x_k\rangle + \frac{3L_{\max}}{2}\|\bar{x}_k - x_k\|^2\right) \\
& \quad +\frac{L_{\text{res}}}{2n}\mathbb{E}\sum_{j \in J(k)}\left(\frac{\alpha}{n}\|\bar{x}_j - x_j\|^2 + \frac{1}{\alpha}\|\bar{x}_k - x_k\|^2\right),
\end{aligned}
$$

where $\alpha > 0$. Letting $\alpha = \sqrt{n}$ we have

$$
\begin{aligned}
T_1 & \leqslant \frac{1}{N}\mathbb{E}\left(\langle \nabla f(\hat{x}_k), \bar{x}_k - x_k\rangle + \frac{3L_{\max}}{2}\|\bar{x}_k - x_k\|^2\right) \\
& \quad +\frac{L_{\text{res}}}{2n^{3/2}}\mathbb{E}\sum_{j \in J(k)}\left(\|\bar{x}_j - x_j\|^2 + \|\bar{x}_k - x_k\|^2\right) \\
& \leqslant \frac{1}{N}\mathbb{E}\left(\langle \nabla f(\hat{x}_k), \bar{x}_k - x_k\rangle + \frac{3L_{\max}}{2}\|\bar{x}_k - x_k\|^2\right) + \frac{L_{\text{res}}\rho^T T}{n^{3/2}}\mathbb{E}\|\bar{x}_k - x_k\|^2.
\end{aligned}
$$

where the last step comes from Lemma 1. Next we verify the requirements in Lemma 1 are satisfied, so that we can safely apply it. If the following holds

$$T(T+1) \leqslant \frac{\sqrt{n}L_{\max}}{4eL_{\text{res}}}$$

$$\gamma = \frac{1}{3L_{\max}},$$

$$\rho := 1 + \frac{4eTL_{\text{res}}}{\sqrt{n}L_{\max}}$$

$$\psi := L_{\max} + \frac{L_{\text{res}}T\rho^T}{\sqrt{n}}\left(2 + \frac{L_{\max}}{\sqrt{n}L_{\text{res}}} + \frac{2T}{n}\right)$$

where the first two are from the requirements of this theorem and the other two are defined here, we have

$$T \leqslant T(T+1) \leqslant \frac{\sqrt{n}L_{\max}}{4eL_{\text{res}}} \tag{17}$$

and

$$\rho^T = \left(1 + \frac{4eTL_{\text{res}}}{\sqrt{n}L_{\max}}\right)^T$$

$$\leqslant \left(1 + \left(\frac{4eL_{\text{res}}}{\sqrt{n}L_{\max}}\right)^{1/2}\right)^{\left(\frac{\sqrt{n}L_{\max}}{4eL_{\text{res}}}\right)^{1/2}}$$

$$\leqslant e. \tag{18}$$

Thus for (14), note that

$$\left(1 - \frac{1}{\rho} - \frac{2}{\sqrt{n}}\right)\frac{\sqrt{n}}{4L_{\text{res}}T\rho^T}$$

$$= \left(\frac{\rho-1}{\rho} - \frac{2}{\sqrt{n}}\right)\frac{\sqrt{n}}{4L_{\text{res}}T\rho^T}$$

$$= \frac{e}{L_{\max}\rho^{T+1}} - \frac{1}{2L_{\text{res}}T\rho^T}.$$

$$\geqslant \frac{1}{L_{\max}} - \frac{1}{2L_{\max}} = \frac{1}{2L_{\max}}.$$

Thus we can verify (14) by

$$\gamma = \frac{1}{3L_{\max}} \leqslant \frac{1}{2L_{\max}} \leqslant \left(1 - \frac{1}{\rho} - \frac{2}{\sqrt{n}}\right)\frac{\sqrt{n}}{4L_{\text{res}}T\rho^T}.$$

Next we verify (12):

$$\rho = 1 + \frac{4eTL_{\text{res}}}{\sqrt{n}L_{\max}} > \left(1 - 2/\sqrt{n}\right)^{-1},$$

by noting that for $n \geqslant 6$

$$1 + \frac{4eTL_{\text{res}}}{\sqrt{n}L_{\max}} \geqslant 1 + \frac{4e}{\sqrt{n}} \geqslant \left(1 - \frac{2}{\sqrt{n}}\right)^{-1}.$$

We verify (13) in the last. Note that

$$\psi = L_{\max} + \frac{L_{\text{res}}T\rho^T}{\sqrt{n}}\left(2 + \frac{L_{\max}}{\sqrt{n}L_{\text{res}}} + \frac{2T}{n}\right)$$

$$\overset{(18)}{\leqslant} L_{\max} + \frac{L_{\text{res}}Te}{\sqrt{n}}\left(2 + \frac{L_{\max}}{\sqrt{n}L_{\text{res}}} + \frac{2T}{n}\right)$$

$$\overset{(17)}{\leqslant} L_{\max} + \frac{L_{\max}}{2}\left(2 + \frac{L_{\max}}{\sqrt{n}L_{\text{res}}} + \frac{L_{\max}}{\sqrt{n}eL_{\text{res}}}\right)$$

$$\leqslant L_{\max} + L_{\max}\left(1 + \frac{1}{\sqrt{n}}\right)$$

$$\leqslant 3L_{\max},$$

where the second last step comes from $L_{\max} \leq L_{\mathrm{res}}$. Thus

$$1/\psi \;\geqslant\; \frac{1}{3L_{\max}} = \gamma.$$

Therefore our choice satisfies the requirements in Lemma 1.

Since the optimatlity condition for

$$\bar{x}_k \;=\; \arg\min_{x\in\Omega}\langle \nabla f(\hat{x}_k), x - x_k\rangle + \frac{1}{2\gamma}\|x - x_k\|^2$$

is

$$\left\langle x - \bar{x}_k, \nabla f(\hat{x}_k) + \frac{1}{\gamma}(\bar{x}_k - x_k)\right\rangle \;\geqslant\; 0, \forall x \in \Omega$$

$$\Rightarrow \langle x - \bar{x}_k, \nabla f(\hat{x}_k)\rangle \;\geqslant\; \frac{1}{\gamma}\|\bar{x}_k - x_k\|^2,$$

we have

$$
\begin{aligned}
T_1 \;&\leqslant\; \frac{1}{N}\mathbb{E}\left(\langle\nabla f(\hat{x}_k), \bar{x}_k - x_k\rangle + \frac{3L_{\max}}{2}\|\bar{x}_k - x_k\|^2\right) + \frac{L_{\mathrm{res}}\rho^T T}{n^{3/2}}\mathbb{E}\|\bar{x}_k - x_k\|^2 \\
&\leqslant\; \frac{1}{N}\mathbb{E}\left(-\frac{1}{\gamma}\|\bar{x}_k - x_k\|^2 + \frac{3L_{\max}}{2}\|\bar{x}_k - x_k\|^2\right) + \frac{L_{\mathrm{res}}\rho^T T}{n^{3/2}}\mathbb{E}\|\bar{x}_k - x_k\|^2 \\
&=\; \left(\frac{1}{N}\left(\frac{3L_{\max}}{2} - \frac{1}{\gamma}\right) + \frac{L_{\mathrm{res}}\rho^T T}{n^{3/2}}\right)\mathbb{E}\|\bar{x}_k - x_k\|^2.
\end{aligned}
$$

Thus putting the upper bound for $T_1$ in, we obtain

$$\mathbb{E}(f(x_{k+1})) - f^* \;=\; \mathbb{E}(f(x_k)) - f^* + \left(\frac{1}{N}\left(\frac{3L_{\max}}{2} - \frac{1}{\gamma}\right) + \frac{L_{\mathrm{res}}\rho^T T}{n^{3/2}}\right)\mathbb{E}\|\bar{x}_k - x_k\|^2 \quad (19)$$

Note that

$$
\begin{aligned}
\left(\frac{1}{N}\left(\frac{3L_{\max}}{2} - \frac{1}{\gamma}\right) + \frac{L_{\mathrm{res}}\rho^T T}{n^{3/2}}\right) \;&\leqslant\; \left(\frac{1}{n}\left(\frac{3L_{\max}}{2} - \frac{1}{\gamma}\right) + \frac{L_{\mathrm{res}}\rho^T T}{n^{3/2}}\right) \\
&=\; \frac{1}{n}\left(\left(\frac{3L_{\max}}{2} - \frac{1}{\gamma}\right) + \frac{L_{\mathrm{res}}\rho^T T}{\sqrt{n}}\right) \\
&\leqslant\; \frac{1}{n}\left(-\frac{3L_{\max}}{2} + \frac{L_{\mathrm{res}}e^{\frac{\sqrt{n}L_{\max}}{4eL_{\mathrm{res}}}}}{\sqrt{n}}\right) \\
&=\; \frac{1}{n}\left(-\frac{3L_{\max}}{2} + \frac{L_{\max}}{4}\right) \\
&\leqslant\; 0.
\end{aligned}
$$

Thus using Lemma 2 and putting $\rho^T \leqslant \mathrm{e}, T^2 \leqslant \frac{\sqrt{n}L_{\max}}{4\mathrm{e}L_{\mathrm{res}}}, T \leqslant \frac{\sqrt{n}L_{\max}}{4\mathrm{e}L_{\mathrm{res}}}, \gamma = \frac{1}{3L_{\max}}$ in (19), we have

$$\mathbb{E}(f(x_{k+1})) - f^*$$

$$\overset{(15)}{\leqslant} \mathbb{E}(f(x_k)) - f^*$$
$$+ \left( \frac{1}{N} \left( \frac{3L_{\max}}{2} - \frac{1}{\gamma} \right) + \frac{L_{\mathrm{res}}\rho^T T}{n^{3/2}} \right) \frac{1}{2\left( \frac{\gamma^2 L_T^2 T^2 \rho^T}{n} + 1 \right)} \mathbb{E}\|\tilde{x}_k - x_k\|^2$$

$$\overset{(17),(18)}{\leqslant} \mathbb{E}(f(x_k)) - f^*$$
$$+ \left( \frac{1}{N} \left( \frac{3L_{\max}}{2} - \frac{1}{\gamma} \right) + \frac{L_{\mathrm{res}}\rho^T T}{n^{3/2}} \right) \frac{1}{2\left( \frac{\frac{1}{9L_{\max}^2}L_T^2 \frac{\sqrt{n}L_{\max}}{4\mathrm{e}L_{\mathrm{res}}}\mathrm{e}}{n} + 1 \right)} \mathbb{E}\|\tilde{x}_k - x_k\|^2$$

$$= \mathbb{E}(f(x_k)) - f^*$$
$$+ \left( \frac{1}{N} \left( \frac{3L_{\max}}{2} - \frac{1}{\gamma} \right) + \frac{L_{\mathrm{res}}\rho^T T}{n^{3/2}} \right) \frac{1}{2\left( \frac{L_T^2}{36L_{\max}L_{\mathrm{res}}\sqrt{n}} + 1 \right)} \mathbb{E}\|\tilde{x}_k - x_k\|^2$$

$$= \mathbb{E}(f(x_k)) - f^*$$
$$+ \left( \frac{1}{N} \left( \frac{3L_{\max}}{2} - \frac{1}{\gamma} \right) + \frac{L_{\mathrm{res}}\rho^T T}{n^{3/2}} \right) \underbrace{\left( \frac{L_T^2}{18L_{\mathrm{res}}L_{\max}\sqrt{n}} + 2 \right)^{-1}}_{b,\text{as defined in Theorem 2}} \mathbb{E}\|\tilde{x}_k - x_k\|^2$$

$$\leqslant \mathbb{E}(f(x_k)) - f^* + \left( \frac{1}{N} \left( \frac{3L_{\max}}{2} - \frac{1}{\gamma} \right) + \frac{L_{\mathrm{res}}\rho^T T}{n^{3/2}} \right) b \frac{2(\mathbb{E}f(x_k) - f^*)}{L\kappa^2} \qquad (20)$$

$$= \left( 1 + \frac{2b}{L\kappa^2} \left( \frac{1}{N} \left( \frac{3L_{\max}}{2} - \frac{1}{\gamma} \right) + \frac{L_{\mathrm{res}}\rho^T T}{n^{3/2}} \right) \right) (\mathbb{E}f(x_k) - f^*)$$

$$\overset{(18),(17)}{\leqslant} \left( 1 - \frac{2b}{L\kappa^2 n\gamma} \left( 1 - \left( 3L_{\max} + \frac{2L_{\mathrm{res}}\mathrm{e}^{\frac{\sqrt{n}L_{\max}}{4\mathrm{e}L_{\mathrm{res}}}}}{\sqrt{n}} \right) \frac{\gamma}{2} \right) \right) (\mathbb{E}f(x_k) - f^*)$$

$$= \left( 1 - \frac{2b}{L\kappa^2 n\gamma} \left( 1 - \frac{7}{4}L_{\max}\gamma \right) \right) (\mathbb{E}f(x_k) - f^*)$$

$$= \left( 1 - \frac{6bL_{\max}}{L\kappa^2 n} \left( 1 - \frac{7}{12} \right) \right) (\mathbb{E}f(x_k) - f^*)$$

$$\leqslant \left( 1 - \frac{2bL_{\max}}{L\kappa^2 n} \right) (\mathbb{E}f(x_k) - f^*),$$

where (20) comes from

$$f(x) - f^* \leqslant \frac{L}{2}\|x - \mathcal{P}_S(x)\|^2$$
$$\overset{(7)}{\Longrightarrow} \frac{2(f(x) - f^*)}{L} \leqslant \kappa^2\|\tilde{x} - x\|^2,$$

completing the proof. $\qquad\qquad\qquad\qquad\qquad\qquad\qquad\qquad\qquad\qquad\qquad\qquad\qquad\qquad$ $\square$