[Reviews · NeurIPS 2016]

Reviewer 1

Summary

This paper studies an asynchronous coordinate descent method with a special sampling scheme. Each processor deals with a set of coordinates and performs greedy updates within this set of coordinates. The algorithm is then implemented and compared to alternatives on support vector machine problems.

Qualitative Assessment

The paper improves the analysis of a previous parallel greedy coordinate descent paper, whose analysis was not enough convincing, with ideas from the asynchronous randomized coordinate descent. I liked the coupling with variable partitioning and would have liked a little more discussion on its impact. However, the idea may be interesting but the paper is not well presented. Asynchronous coordinate descent is not an easy subject and the authors write their paper as if the reader were experts of Lui et al's paper. In particular, they use non-standard notation before defining it. I also have a few minor issues listed below. Line 42: THE optimization community Line 55: although the worst case rates are the same: the paper "Nutini, Schmidt, Laradji, Friedlander, Koepke. Coordinate descent converges faster with the Gauss-Southwell rule than random selection" showed a better rate. Line 75: in contract -> in contrast Line 80: each machine asynchronously communicate -> each machine asynchronously communicates Line 107: The notation with hats and without hats is quite tricky and fundamental to the proofs. Assumption 1 (which defined the notation) should be given before Eq (2) and Algorithm 1. Algorithm 1, Step 4: what is \hat x ? Algorithm 1: it is not at all clear that the algorithm is run on several processors in parallel. Line 126: Please remove "both" Line 138: what is x^2? Line 139: some general convex objectives -> some weakly convex objectives Line 150: T -> T^2 Line 176: g is used before its definition Algorithm 2, Lines 5 and 6: i's are missing (idem for Algorithm 3) Algorithm 3, Line 3: argmax_i -> argmax_{i \in S_t} Line 260: Fig. 1 (d)-(f) are not the good ones. You put the ones from Fig. 3 (a)-(c) instead. Line 370: On the fourth inequality, you are doing as if J(k) were not random, while it obviously is. Line 379: the equality is an inequality for alpha = sqrt(n). As a final remark, it would have been better to prove some theoretical improvement against PSCD.

Confidence in this Review

2-Confident (read it all; understood it all reasonably well)


Reviewer 2

Summary

This paper considers asynchronous parallel greedy coordinate descent (Asy-GCD) algorithm for minimizing a smooth function with decomposed constraints. The algorithm works in a parameter server framework, with each worker performing greedy CD asynchronous. This is different from [17] which picks the coordinate stochastically. Efficient coordinate search algorithm is proposed for kernel SVM by maintaining the gradient, which is updated by atomic writes. The results are mainly two folds. First in theory, a linear convergence rate is established. Second in practice, by using kernel caching and sensible variable partitioning, the proposed algorithm outperforms existing solvers in a shared-memory multi-core setting.

Qualitative Assessment

This paper is well written and the innovations and improvements can be seen clearly. Although changing stochastic into greedy is not that big innovation and the O(l) algorithm for coordinate selection based on maintaining the gradient is also not new, I can kind of appreciate the practical improvement after putting together some existing tricks (Sec 4.3). My major concern is, however, the rates of convergence. There are five “constants” involved in Theorem 2, \kappa, L_res, L_max, L, and L_T. It is critical to unfold their values for concrete problems, e.g. the kernel SVM problem considered in the paper. Linear rates are useful only if its condition number depends on the dimensions of the problem mildly. What is most unsettling is the L_T^2 term in b. How does L_T depend on T? If it is \Omega(\sqrt{T}), then this will translate to an iteration complexity that is linear in T. Since “T is usually proportional to the total number of cores involved in the computation”, this will cancel the speedup proffered by parallelization and asynchronous updates. It is therefore crucial that the paper clarify the value of these “constants” explicitly. =================== Post-rebuttal: I read the rebuttal, and now feel less worried about the dependence on L_T (or T). However, based on the numbers provided in the rebuttal, now I am rather worried about the constants \kappa and L. Using the rebuttal's argument that b is almost a universal constant, the rate of convergence is roughly L n \kappa^2 \log (1/\epsilon) The values of L and \kappa provided in the rebuttal seem to grow linearly in n. This is unsettling because such a high order dependency on n essentially leaves the linear rate in vain. The submission allows parallelization up to n^{1/4} cores. So roughly speaking, the cost is O(n^{15/4} \log (1/\epsilon)) 15/4 might be over-estimated a little, but it is far worse than the following rate for training nonlinear kernel SVM given by [a]: O(n / \epsilon). The disadvantage is even more evident considering that \epsilon rarely needs to be less than 1/n (consider n = 4*10^5 in the experiment where #example is denoted as \ell instead of n). Having said that, I think think this paper actually has some high quality material. The experiments are surely good. The linear rates were first shown for this problem, although further refinement is needed on its condition number. Overall, this paper is above the acceptance threshold of NIPS, and I've updated my ratings accordingly. [a] Bernd Gärtner and Martin Jaggi, Coresets for Polytope Distance, Symposium on Computational Geometry, 2009.

Confidence in this Review

3-Expert (read the paper in detail, know the area, quite certain of my opinion)


Reviewer 3

Summary

This work proposes variant of parallel coordinate descent in the shared-memory setting. On each iteration each processor samples a block from a among a fixed set of blocks, and then selects a variable to update from within the block based on the Gauss-Southwell rule. The algorithm is notably asynchronous, and the algorithms show a linear convergence rate for the method for problems satisfying certain error bounds. A small number of experiments were performed.

Qualitative Assessment

I've read the author response and agree with the points raised there. I'm raising my scores but I'm too lazy to update my original review (see below) but still think you update the paper to comment on these issues. --- It is a good idea to combine asynchronous parallel updates with greedy coordinate selection and this combination is novel. However, there are a lot of closely-related works in the recent literature (parallel randomized coordinate descent, greedy coordinate descent methods, asynchronous versions of coordinate descent methods) and for me this work didn't really give any new insights that make it stand out among them. One thing that did surprise me was that Theorem 2 uses a standard O(1/L) step-size which is nice because it doesn't depend on quantities that are difficult to obtain. However, the theorem only applies under condition (8) which seems to be quite difficult to compute. Indeed, if something like this condition is not satisfied then I think the method would not converge. Is the constant in Theorem 2 necessarily less than 1? I believe the analysis here suffers from the issue pointed out in the recent ASAGA work of Leblond et al., so it would be good for the authors to look into this. A related work is that of Nutini et al. from ICML last year. That work shows faster convergence rates for greedy (e.g., Gauss-Southwell) variable selection. It also discusses why the projected-gradient "GS-r" rule used in this work may not be ideal for proximal-gradient settings. The key disadvantage of the greedy approaches is that they require either O(l^2) memory or a high computational cost to compute the greedy rule, making it hard to apply this method to very large problems. This is briefly mentioned discussed in 4.3, but I think this crucial implementation issue deserves some more discussion. Unless I'm not understanding correctly, the algorithm is not fully-asynchronous because of the locking required to update g after an update to alpha. It's nice that the authors include the time for running k-means as part of the experiment. The labels of the figures in Section 5.1 seem to be mixed up. The experiments are a bit underwhelming in that LIBSVM seems to do similarly to the proposed algorithm on two out of three datasets. These are also fairly small datasets.

Confidence in this Review

3-Expert (read the paper in detail, know the area, quite certain of my opinion)


Reviewer 4

Summary

The authors suggest a new optimization algorithm, Asynchronous Parallel Greedy Coordinate Descent, which is an asynchronous version of the greedy coordinate descent algorithm. They show that, under some conditions, the algorithm converges with a linear rate.

Qualitative Assessment

This seems like a solid analysis of a pretty useful algorithm that might actually be better in practice than state-of-the-art techniques (the experiments definitely illustrate that this is the case). Some specific questions. 1) How does your rate of convergence compare to that of other competing algorithms? In particular, how does it compare to non-asynchronous greedy coordinate descent?

Confidence in this Review

2-Confident (read it all; understood it all reasonably well)


Reviewer 5

Summary

The paper introduces version of Greedy Coordinate Descent, that is suitable for asynchronous parallel implementation. Analysis is done in fairly general setting, followed by clear explanation how to one can implement such method efficiently for kernel SVMs. Recommendation: Accept (clear accept, but not near oral+ level)

Qualitative Assessment

POST REBUTTAL EDITS: No major change in my assessment after rebuttal. Skimming through the paper again, I would maybe set all grades as 3, and still recommend accept. ORIGINAL REVIEW: I think the paper is very well structured, all important concepts well explained, and consequently easy to read and understand. I managed to understand implementation details relatively quickly even though I never really worked with SVM implementation. The clarity and level of language is very good overall, but seems to be lower in Section 5 - possibly finished in hurry before deadline. I feel this will be improved to match the earlier sections for the final version. I find the algorithm could be also interpreted as an interesting mix between RCD and GCD, due to the random selection of sets S_k. Phrasing like this could get more attention from the community. I spent a while looking at the proofs, but did not check every step in detail. I am not sure about Technical quality and Novelty/originality grades, as I don't exactly know under what assumptions were recent related algorithms analysed. For instance, how many analyses need to assume consistent read, and similar. Both grades should be 3 or 4. I am leaning towards 3, as the topic is not exactly the main focus of NIPS. Specific remarks: I don't have any major comments, only details to polish. - Section 2: missing spaces after subtitles - #107: (eq (2)) I would mention \hat{x} here or earlier. I was confused for a while about what that is. - Algorithm 1: please remove semicolons - #151: cores - End of Section 3: I would ideally also compare with serial BCD, to highlight the effect of parallelism and asynchrony. - Reference you miss but seems very relevant: Julie Nutini, Mark Schmidt, Issam H Laradji, Michael Friedlander, Hoyt Koepke: Coordinate Descent Converges Faster with the Gauss-Southwell Rule Than Random Selection (ICML 2015) - Algorithm 2: I would prefer the comment (See eq (2)) flushed to right border, perhaps without the "eq". Also, having numbered lines, and starting additional manual numbering from line 3 seems weird. - Figure 1 (d-f): I believe you have wrong plots there. - #262: please remove "perfect". It is not perfect. It is very good. But even if it was exactly linear with derivative 1, I wouldn't call it perfect, as it could be superlinear. - Figures: make the column ordering consistent across figures. - Figure 2: Maybe having bottom left corner (1,1) would be better. Also, in (a) it seems like you miss one point.

Confidence in this Review

2-Confident (read it all; understood it all reasonably well)


Reviewer 6

Summary

This paper proposed an asynchronous parallel coordinate descent algorithm for solving convex optimization problems. Under several assumptions, the algorithm converges lineary to the optimal solution.

Qualitative Assessment

1. How can we make sure that 'Global error bound' assumption is satisfied for an optimization problem (not necessarily SVM)?

Confidence in this Review

2-Confident (read it all; understood it all reasonably well)